# Personal and Family Resources Related to Depressive and Anxiety Symptoms in Women during Puerperium

**DOI:** 10.3390/ijerph17145230

**Published:** 2020-07-20

**Authors:** David Feligreras-Alcalá, Antonio Frías-Osuna, Rafael del-Pino-Casado

**Affiliations:** Department of Nursing, Faculty of Health Sciences, University of Jaén, 23071 Jaén, Spain; dfa00001@red.ujaen.es (D.F.-A.); rdelpino@ujaen.es (R.d.-P.-C.)

**Keywords:** depression, postpartum, anxiety, risk factors, puerperium, pregnancy

## Abstract

*Introduction*: This study investigated the relationship between personal and family resources (i.e., social support, family functioning, self-efficacy in care, sense of coherence and perceived burden of care) and depressive and anxiety symptoms in women during the puerperium, adjusting for stressors. *Methods*: This is a quantitative research design, carried out through a descriptive, cross-sectional correlation study. This study includes 212 women over the age of 19 who gave birth from March to September 2019 in Maternal and Child Hospital of Jaén (Spain). Women were selected during the immediate postpartum period. The variables analysed were postpartum depressive symptoms (Edinburgh scale), anxiety symptoms (STAI state anxiety questionnaire), perceived social support (Duke-UNC-11), family functioning (family APGAR), self-efficacy in care (Lawton), sense of coherence (SOC-13), perceived burden (Caregiver Strain Index) and stressful life events (Holmes and Rahe). The main analysis consisted of a multiple linear regression. *Results*: The regression model of depressive symptoms found a positive association with perceived burden (β = 0.230, *p* = 0.015) and negative associations with self-efficacy in care (β = −0.348, *p* < 0.001), social support (β = −0.161, *p* < 0.001) and sense of coherence (β = −0.081, *p* = 0.001). The regression model of anxiety symptoms obtained a positive association with perceived burden (β = 1.052, *p* < 0.001) and negative associations with self-efficacy in care (β = −0.329, *p* = 0.041), social support (β = −0.234, *p* = 0.001) and sense of coherence (β = −0.262, *p* < 0.001). *Discussion*: Firstly, depressive and anxiety symptoms in the puerperium period may be more prevalent than in other periods of a woman’s life. Secondly, perceived social support, self-efficacy in caring for the newborn and sense of coherence may be protective factors for depressive and anxiety symptoms in the puerperium period. Finally, perceived burden in caring for the newborn may be a risk factor for these symptoms.

## 1. Introduction

The postpartum or puerperium period is defined as the period of 6–8 weeks beginning 1 h after the birth of the foetus and the expulsion of the placenta, and it reflects the approximate time needed for uterine involution and return of most maternal organic systems to their pre-pregnancy state; it is also characterised by psychosocial adaptations, including changes in parental role [1,2], modifications in family relationships [3] and alterations in self-perception and body image [4]. It represents a period of special vulnerability for women in the appearance of different psychological disorders, especially depressive symptoms [1] and anxiety [5].

Depression is a mental disorder that constitutes the main cause of disability in the world and contributes significantly to the overall global burden of morbidity [6]. The lifetime rate of depression is around 20% of the population, affecting more women than men [7]. Puerperal depression is not recognised as a separate diagnosis of depression in the Diagnostic and Statistical Manual of Mental Disorders (DSM). However, in its fifth edition (DSM-V), it includes as a novelty compared to its previous version, a major depressive disorder with onset in the peripartum, considering its development during pregnancy or in the first four weeks of postpartum [8]. The World Health Organization (WHO), in its 11th Revision of International Statistical Classification of Diseases and Related Health Problems, defines puerperal depression as a mild mental and behavioural disorder that begins in the 6 weeks of postpartum [6].

Depressive disorders during the postpartum period can be classified into three categories: postpartum sadness, postpartum psychosis and postpartum depression. Postpartum sadness affects 50–85% of women, appearing around the 4th day and disappearing on the 10th postpartum day, with symptoms such as short periods of crying, anxiety, sadness, difficulty sleeping, confusion and irritability. Postpartum psychosis has a prevalence of 0.1–0.2%, occurring in the first 2 weeks postpartum. It includes symptoms such as agitation, paranoia, disorganised thinking and hallucinations. It requires emergency treatment for risk of suicide or harm to the newborn [9,10]. 

Postpartum or puerperal depression has a prevalence of 12–13% at 6 weeks of postpartum in industrialised countries [1,11], as well as a prevalence period of 19% in the first 12 months of postpartum [12]. In Spain, the studies carried out place it between 10% and 13% [13,14]. The usual clinical manifestations are depressive mood, irritability, loss of interest in habitual activities, insomnia, fatigue and loss of appetite [10]. The consequences of depressive symptoms during puerperium stage not only affect the social and occupational functioning of women but also extend to the couple, the family and mother–child interaction [15]. Puerperal depression is associated with a worse quality of life for the newborn, which may affect their emotional, intellectual and cognitive development [16,17].

The prevalence of puerperal depression can be determined through questionnaires, used as a screening method for depressive symptoms in this period, or through a structured clinical interview, which determines the diagnosis of major or minor depressive disorders [9]. The prevalence of a major depressive episode diagnosed through clinical interview is lower than the prevalence established through screening [18].

Anxiety is a common disorder in women, since 30% of them experience at least one episode throughout their lives [7]. During the puerperium, prevalence of anxiety is 13–18% [19,20], and the combined presence of both disorders, anxiety and depression is estimated at 13% [20]. The presence of anxiety in puerperium is related to different adverse outcomes, such as a decreased effectiveness of the parental role, difficulty in coping or lower levels of confidence [21,22,23].

Despite negative consequences of the presence of depressive symptoms or anxiety on women’s health in the postpartum period, this is a neglected aspect by the health services, as well as the object of few interventions in prevention and promotion of health [24]. These interventions inevitably go through identification of risk factors of both processes.

Traditionally, literature has studied different factors associated with the appearance of depressive symptoms or anxiety in the postpartum period, which could be divided based on different areas of women: physical or biological factors, such as delicate physical health [25]; psychological factors, such as depression or anxiety during pregnancy [26]; previous psychiatric illness [26] or a poor relationship with her couple [13]; obstetric factors, such as unwanted pregnancy [27]; paediatric factors, such as initiation or not of breastfeeding [28]; or socio-demographic factors, such as maternal age [29] or low economic status [26].

In addition, the consequences of the presence of anxiety and postpartum depression on emotional regulation, the formation of the attachment system and the development of the child are currently gaining importance; several studies show that the presence of anxiety and postpartum depression is associated with a lower performance of infant language, as well lower social engagement, less mature regulatory behaviours and more negative emotionally, and higher cortisol reactivity [30,31]. In this sense, in the systematic review carried out by Warfa et al. the presence of insecure attachment appears as an important risk factor in the development of postpartum depression, as well as its relationship with other factors such as low social support or a lower level of family functioning [32]. Likewise, there are also factors associated with the presence of depressive symptoms or anxiety in the postpartum period that have been studied to a lesser extent than those described above and that we could include in a personal and family resources model. Among these factors, the authors find social support (i.e., instrumental, emotional or informational support provided by family and friends) [33], family functioning (i.e., perception of the level of fulfilment of basic family functions) [34], self-efficacy (i.e., belief of efficacy) in care [35], sense of coherence [36] and perception of the burden of care [37]. Sense of coherence is defined as a global orientation that expresses the degree of confidence that the stimuli that come from internal and external environments of oneself in the course of life are structured, those are predictable and explainable, that resources are available to meet the demands represented by these stimuli and these demands are challenges, worthy of investment and commitment [36]. Perceived burden is defined as the caregiver’s assessment that the care situation is beyond their ability to cope with adequately [37].

Although previous personal and family resources have been related to emotional health in other areas of family care [38,39], their study in puerperium has been scarce and inconclusive with only a few studies that analyse issues related to personal and family resources [40,41,42,43,44,45,46,47,48]. Therefore, further research is needed on this issue. 

With the development of this research, the authors have proposed the following objectives: (1) to determine point prevalence of depressive symptoms and anxiety in women at 6 weeks after delivery; (2) to establish the possible relationship between personal and family resources (i.e., perceived social support, family functioning, self-efficacy in care, sense of coherence and perceived burden) and depressive symptoms and anxiety in women in puerperium; and (3) to identify a possible risk profile for the development of postpartum depression and anxiety in women during puerperium.

## 2. Methods

### 2.1. Design

A quantitative research was carried out through a descriptive, cross-sectional design of correlation in women during puerperium and located in the province of Jaén, Spain.

### 2.2. Ethical Considerations

This study was approved by the Research Ethics Committee of Jaén (Public Health System of Andalusia). This study was carried out in accordance with state legislation and the principles established in the Helsinki Declaration of 1964. Informed Consent was requested as a guarantee of respect for bioethical principle of autonomy. The confidentiality and privacy of data obtained was guaranteed. Participants were given the opportunity, both verbally and in writing, to refuse to participate or leave the study at any time.

### 2.3. Participants

The participants in this study were women who gave birth at Maternal and Child Hospital of Jaén between March and September 2019. After an exhaustive review of literature, the authors determined to carry out a systematic random sampling. The sample size was calculated using Epidat 3.1 program, assuming a significance level of 5% and a power of 80%, in which 200 sample units are necessary to detect a correlation coefficient of at least 0.175 [49]. This sample size was increased by 10% to prevent future non-responses. Considering the number of births estimated at Maternal and Child Hospital of Jaén during the study period, in addition to the necessary sample of 220 women, a sampling fraction of five was established. The first sample was selected by lottery method with simple random sampling. Every following sample was selected from the Hospital’s birth register until the desired sample size was obtained. The exclusion criteria were the following: (1) equal age or less than 19 years, (2) previous history or current staff of psychiatric pathology, (3) serious illness or exitus of the newborn, (4) not understanding the Spanish language, (5) not accepting participation in the study or (6) not signing the informed consent.

### 2.4. Study Measures

#### 2.4.1. Sociodemographic Variables

Some of the variables collected were marital status, educational level, employment status, family income, pregnancy search and family history of psychiatric pathology.

#### 2.4.2. Independent Variables

The independent variables of this study are the perceived social support, perception of family functioning, self-efficacy in newborn’s care, sense of coherence and perceived burden.

Perceived social support was measured by the Duke-UNC-11 questionnaire [33], a self-administered instrument that measures social support both in its affective dimension (i.e., referring to expressions of love, appreciation, sympathy or belonging) and its confidential dimension (i.e., that through which people can receive information, advice or guidance). It consists of 11 items along with a 5-point, Likert response scale, ranging from 1 (Much less than I want) to 5 (As much as I want). Scoring for total questionnaire ranges from 11 to 55 points (directly proportional to the level of perceived social support). This questionnaire has been validated in the Spanish population with adequate clinimetric properties (e.g., Cronbach’s alpha coefficient of 0.93) [50]. In this study, Cronbach’s alpha coefficient was 0.91.

The perception of family functioning was assessed using the family APGAR questionnaire [34], which is a self-administered questionnaire that shows how a person perceives the level of functioning of a family unit. It consists of five questions, each of which can take one of the following scores: 0 (Almost never), 1 (Sometimes), 2 (Almost always). The range of scores is between 0 and 10 points. This questionnaire is validated in the Spanish population, presenting a Cronbach alpha coefficient of 0.84 [51]. In our study, Cronbach’s alpha coefficient was 0.85.

Self-efficacy in newborn care was measured by the mastery or self-efficacy dimension of Lawton’s questionnaire [35]. This is a self-administered questionnaire that includes 12 items with a Likert response scale, ranging from 1 (totally disagree) to 5 (totally agree), where higher scores indicate greater self-efficacy in care. The items on this scale were adapted for newborn care in this study. In this study, Cronbach’s alpha coefficient was 0.77.

The sense of coherence was collected using the SOC-13 Sense of Coherence scale [36]. This scale assesses three dimensions of sense of coherence: compressibility, manageability and significance, which are closely related. It consists of 13 items that are answered on a Likert scale with seven scores, ranging from 1 (always) to 7 (never), where a higher score indicates a greater sense of coherence. This scale is validated in the Spanish population, presenting a Cronbach’s alpha of 0.80 [52]. In this study, Cronbach’s alpha coefficient was 0.88.

Perceived burden was assessed using the Caregiver Strain Index [53], a scale that measures the degree of overexertion of carers of dependent people in general. It consists of 13 items with a true or false dichotomous response. Each affirmative answer scores 1, so the range of scores goes from 0 to 13 (directly proportional to burden). This instrument was validated in the Spanish caregiver population [54], with a Cronbach’s alpha of 0.80. The items of this scale were adapted for newborn’s care in the performance of this study. In this study, Cronbach’s alpha coefficient was 0.71.

#### 2.4.3. Dependent Variables

Dependent symptoms and anxiety in puerperium were included as dependent variables.

Depressive symptoms were measured using the Edinburgh postpartum depression scale [55]. This scale is used to detect depressive states in the postpartum period. It is a self-administered scale of 10 items, with four possible response alternatives, scored from 0 to 3, depending on the severity of symptoms. Scores range from 0 to 30 points (proportional to level of depressive symptoms). A cut-off point, equal to or greater than 10 points, is considered adequate to detect depressive symptoms in this period, with a sensitivity of 79%, a specificity of 95% and a positive predictive value of 63% [14]. Its use is recommended in the first 6 weeks of postpartum to ensure correct screening for depressive symptoms in puerperium, according to the Ministry of Health, Social Services and Equality of Spain [56]. It is validated and widely applied in Spain [14]. In this study, Cronbach’s alpha coefficient was 0.86.

Anxiety was measured by STAI state-trait’s anxiety questionnaire [57], a self-administered instrument that measures two independent concepts of anxiety: on the one hand, anxiety as a state is referred to as a transient emotional condition; on the other hand, anxiety as a trait is described as a relatively stable anxious propensity. In the development of this research, anxiety-status was evaluated as a specific measure of anxiety during this study period. This subscale consists of 20 items, with 4-point, Likert responses (proportional to intensity of anxiety). Total subscale scores range from 0 to 60 points, with the 75th percentile recommended as a cut-off point in adult women [57]. This questionnaire is validated in the Spanish population, presenting a Cronbach’s alpha coefficient of 0.94 [58]. In this study, the Cronbach’s alpha coefficient of the state subscale was 0.94.

#### 2.4.4. Control Variable

The presence of stressful life events was included as a control variable, and these were measured using the Holmes–Rahe social readjustment scale [59]. It consists of 43 vital events to which a score is assigned according to the stress it generates in the person. Stressful events experienced in the last year are selected and previously established scores are added. This scale was adapted to the Spanish population [60].

### 2.5. Data Collection Procedure

Women were captured during their hospital admission after childbirth, in the period between 15 March 2019 and 15 September 2019. The study was explained to those selected women and they were offered to participate in the study. Each woman who agreed to participate was given the Informed Consent model for reading and signing and corresponding questionnaires. These questionnaires were self-filled anonymously during the 6th week after the delivery. There was a total of eight non-responses (3.64%), so a definitive sample was 212 women.

### 2.6. Data Analysis

A descriptive analysis of the quantitative variables was carried out through measures of central tendency (i.e., means or medians) and dispersion measures (i.e., standard deviations or interquartile range), in addition to the frequencies and percentages of the qualitative variables. A correlation matrix was performed to explore the relationships between the variables, and the Pearson’s correlation coefficient (*r*) and the Spearman’s *Rho* coefficient were used (in those variables that did not meet the normality assumptions). For the multivariable analysis, a multiple linear regression (stepwise method) was carried out, with prior confirmation of assumptions of the linear regression model: normality (Kolmogorov–Smirnov test); linearity (seen with partial regression graphs); homoscedasticity (graph of standardised residue); error independence (Durbin-Watson); non-collinearity (variance inflation factor and tolerance). Those variables that did not meet normality were transformed by Box-Cox transformations, with the help of Statistica 8.0 (Stat Soft Inc., Tulsa, USA) program [61]. For the correlations and multivariable analyses, the level of statistical significance was set at 0.05. Relationships between the proposed variables were examined with the help of the SPSS v22.0 (IBM International Business Machines Corporation, Armonk, NY, USA) program.

## 3. Results

### 3.1. Description of Sample and Study Variables. Prevalence of Depression and Anxiety

Descriptive data of the sample and study variables are shown in Table 1. Out of the total of 220 postnatal women sampled, 212 participated in the study; making a response rate of 96.36%. The average age of women who participated is 32 years, with a minimum age of 19 years and a maximum age of 47 years. The majority of the women were married (78.3%) and were in active employment status on their own or another’s account (69.4%). In addition, 47.6% of participants had university studies, and 90.1% of women reported that pregnancy had been sought.

The average score of depressive symptoms is 7.20 (95% CI; 6.53–7.89). Using the cut-off point of 10 in the EPDS questionnaire, the point prevalence of depressive symptoms at 6 weeks after delivery is 26.9% (95% CI; 21.2–33.0). The average anxiety score is 17.26 (95% CI; 15.95–18.67). Taking the 75th percentile, the point prevalence of anxiety at 6 weeks postpartum is 27.8% (95% CI; 21.7–34.0). Joint prevalence in our sample of comorbidity of both disorders is 20.3% (95% CI; 15.1–25-5).

### 3.2. Factors Related to Depression and Anxiety

The variables that met normal assumptions were the following: depressive symptoms, anxiety, sense of coherence, self-efficacy in care and perceived burden. The variables that did not meet normal assumptions, however, were social support, family functioning and stressful life events; therefore, Box-Cox transformations were carried out. No breaches were detected in the rest of the assumptions in regression model (e.g., Durbin-Watson of 1.94 for depressive symptoms and 2.02 for anxiety; and tolerances between 0.45 and 0.86 for depressive symptoms and between 0.45 and 0.85 for anxiety).

Regarding correlations (Table 2) a positive association was found between the presence of depressive symptoms and perceived burden (r = 0.429; *p* < 0.001) and stressful life events (rho = 0.240; *p* = 0.001). In addition, a negative association was found between depressive symptoms and family functioning (rho = −0.364; *p* < 0.001), social support (rho = −0.537; *p* < 0.001), sense of coherence (r = −0.616; *p* < 0.001) and self-efficacy in care (r = −0.466; *p* < 0.001). Regarding anxiety, a positive association was found with perceived burden (r = 0.532; *p* = 0.01) and stressful life events (rho = 0.186; *p* = 0.001). Likewise, a negative association was found between anxiety and family functioning (rho = −0.370; *p* < 0.001), social support (rho = −0.539; *p* < 0.001), sense of coherence (r = −0.644; *p* < 0.001) and self-efficacy in care (r = −0.361; *p* < 0.001).

In the regression model of depressive symptoms, independent variables were included together with the control variable, stressful life events. The resulting model has an *R^2^* of 0.52 (Table 3). In this model, the presence of depressive symptoms is positively associated with the perceived burden (β = 0.133; *p* = 0.015) and negatively with self-efficacy in care (β = −0.253; *p* < 0.001), social support (β = −0.334; *p* < 0.001) and sense of coherence (β = −0.231; *p* < 0.001). No association was found with family functioning or stressful life events. Regarding the anxiety regression model, it has an *R^2^* of 0.58 (Table 4). In this model, anxiety is positively associated with the perceived burden (β = 0.273; *p* < 0.001) and negatively with self-efficacy in care (β = −0.107; *p* = 0.041), social support (β = −0.217; *p* = 0.001) and the sense of coherence (β = −0.333, *p* < 0.001). No association was found with family functioning or stressful life events.

## 4. Discussion

In this study, approximately one in four women had depressive symptoms at 6 weeks postpartum—the proportion of anxiety being similar. These figures show the importance of these problems, which are higher than those found in other periods outside the puerperium [1,12,21,28,62]. The results are superior to those of other studies in the postpartum period [20,63,64]. The differences could be due to the fact that the questionnaires used are screening tools, so they can overestimate the presence of the disorders [65], the cut-off point used in the questionnaires [65] or even the moment at which the prevalence is measured (the figures decrease as the postpartum) [66].

In this study, the authors found that less perceived social support, greater perceived burden, less self-efficacy in care and a lesser sense of coherence are accompanied by increased depressive symptoms and anxiety in women in the puerperium (i.e., at 6 weeks after delivery), regardless of the presence of other stressful life events unrelated to the care of the newborn.

Our results are consistent with those of other studies that have analysed the relationship between anxiety, depression and social support in postpartum women [41,67,68,69,70], self-efficacy in care [42,71,72] and sense of coherence [40,73,74]. The authors have not found studies that relate anxiety and depression to perceived burden, although there are studies that relate the above problems to perceived stress in relation to newborn care [43,75,76], where they show a directly proportional relationship. Perceived burden can be considered as the assessment of the care situation as stressful [77].

The findings of this study improve the available evidence on the possible risk factor role of perceived burden on anxiety and depression in the puerperium and the possible protective effect of perceived social support, self-efficacy in care and the sense of coherence. This is because in this study the effect of these factors was analysed together, controlling for possible confounding factors and with a sufficient sample size and extracted by probabilistic sampling, with a very low percentage of refusal to participate in the study. In addition, this study highlights the important contribution of the above factors in explaining the variance of depressive symptoms and anxiety.

According to Lazarus and Folkman [78], the proliferation of stress towards its negative emotional consequences, among which are anxiety and depression, occurs when assessing potentially stressful situations as a threat to which the person cannot respond. The data support the hypothesis that those women who perceive as adequate or satisfactory the support they receive from family and friends in the care of the newborn, perceive their capacities regarding such care as adequate, and have a greater sense of coherence, so they see the care situation as understandable (i.e., internal and external stimuli are perceived as ordered, structured and rationally understandable), manageable (i.e., they believe they have the necessary resources to cope with it) and significant (i.e., they feel it is worth dedicating efforts to it), valuing the care situation as less stressful, in line with the theoretical approaches of Cohen et al. [79] for social support, Bandura [80] for self-efficacy and Antonovsky [36] for the sense of coherence. On the other hand, our results support the hypothesis that those women who perceive a higher level of perceived burden in the realisation of care related to the newborn, could feel overcome by the care situation, trapped by it and excluded from their social relationships and that this perception can make them perceive the care situation as more stressful [77].

The findings of this study allow us to propose a risk profile of anxiety and depression in the puerperium that would be characterised by low perceived social support, high perceived burden, low self-efficacy in care and low sense of coherence. This risk profile could be useful for screening and early detection of women at high risk of anxiety and depression. On the other hand, the results justify assessing the feasibility of interventions aimed at improving the perception of competence in care [81], to work on the sensation of feeling connected as a way to improve the perception of social support received [82], to promote the sense of coherence [39] and to alleviate the feeling of burden in care by fostering a positive perception of the newborn rearing [83].

## 5. Limitations

The study has some limitations that should be noted. Firstly, it is a cross-sectional, descriptive study of correlation, so it can only describe the associations between the study factors, and the causality between them cannot be deduced. In addition, because of the foregoing and with regard to the determination of prevalence, we lack information on whether the prevalence found vary when approaching or moving away at the time of delivery. Secondly, the questionnaires used are screening tools, so they can overestimate the prevalence of anxiety and depression. This is the reason why, in this study, the authors talk about depressive symptoms and anxiety and not depression and anxiety, respectively.

## 6. Conclusions

Despite the above limitations, we can reach the following conclusions: (1) depressive and anxiety symptoms in the puerperium period may be more prevalent than in other periods of a woman’s life; (2) perceived social support, self-efficacy in caring for the newborn and sense of coherence can be protective factors for depressive symptoms and anxiety in the puerperium; (3) perceived burden in caring for the newborn may be a risk factor for the mentioned symptoms.

At the level of clinical practice, screening for the above factors may be justified for early prevention of depressive and anxiety disorders in the postpartum period.

## Figures and Tables

**Table 1 ijerph-17-05230-t001:** Description of studied sample.

**(I)**			
**Variables**	***n*** **(%)**	**M (SD)**	**Median (IQR)**
Age		32.67 (4.58)	
Marital Status			
Single	8 (3.80)		
Married	166 (78.30)		
With couple	35 (16.50)		
Separated or divorced	3 (1.40)		
Education level			
Primary	18 (8.50)		
Secondary	21 (9.90)		
High School	12 (5.70)		
FP Middle degree	36 (17.00)		
FP Higher degree	24 (11.30)		
University	101 (47.60)		
Employment situation			
Student	3 (1.40)		
Active or own account	39 (18.40)		
Asset	108 (50.90)		
Unemployed	47 (22.20)		
Domestic work	15 (7.10)		
Family income			
< from 500€	1 (0.50)		
From 500 to <1000€	27 (12.70)		
From 1000 to <1500€	60 (28.30)		
From 1500 to <2000€	54 (25.50)		
From 2000 to <2500€	29 (13.70)		
From 2500 to <3000€	28 (13.20)		
From 3000 to <5000€	10 (4.70)		
> from 5000€	3 (1.40)		
Pregnancy wanted			
Yes	191 (90.10)		
No	21 (9.90)		
Number of pregnancies		1.835 (0.98)	
Mode of delivery			
Eutocic	141 (66.50)		
Instrumental	32 (15.10)		
Caesarean section	39 (18.40)		
Sex of newborn			
Man	112 (52.80)		
Woman	100 (47.20)		

**(II)**			
**Variables**	***n*** **(%)**	**M (DE)**	**Median (IQR)**
Family history of psychiatric pathology			
Yes	13 (6.10)		
No	199 (93.60)		
Depressive symptoms prevalence	57 (26.90)		
Anxiety		17.255 (10.64)	
Anxiety prevalence	59 (27.80)		
Comorbid prevalence of depressive and anxiety symptoms	43 (20.30)		
Family functioning			9 (8.00–10.00)
Social support			47.50 (37.00–52.00)
Sense of coherence		67.066 (13.50)	
Self-efficacy in care		23.943 (3.46)	
Perceived burden		5.231 (2.76)	
Stressful life events			181 (150.00–248.50)
Source: self-made			

**Table 2 ijerph-17-05230-t002:** Correlation matrix of measures of study variables.

Variables	1	2	3	4	5	6	7	8	9	10
1 Depressive symptoms	1									
2 Anxiety	0.772 **	1								
3 Family functioning	−0.364 **	−0.370 **	1							
4 Social support	−0.537 **	−0.539 **	0.570 **	1						
5 Sense of coherence	−0.616 **	−0.644 **	0.587 **	0.689 **	1					
6 Self-efficacy in care	−0.466 **	−0.361 **	0.152 *	0.273 **	0.376 **	1				
7 Perceived burden	0.429 **	0.532 **	−0.314 **	−0.379 **	−0.447 **	−0.258 **	1			
8 Stressful life events	0.240 **	0.186 **	−0.260 **	−0.236 **	−0.305 **	−0.050	0.211 **	1		
9 Age	0.032	0.101	−0.019	−0.052	−0.035	−0.107	0.075	−0.109	1	
10 Number of pregnancies	0.075	0.097	0.005	−0.102	-0.072	0.017	0.033	−0.010	0.318 **	1

** Correlation is significant at 0.01 level; * correlation is significant at 0.05 level; source: self made.

**Table 3 ijerph-17-05230-t003:** Multiple linear regression by steps (dependent variable: depressive symptoms).

Variables	Beta	Standard Error	Beta Standardised	*p*
Perceived burden	0.230	0.094	0.133	0.015
Self-efficacy in care	−0.348	0.072	−0.253	<0.001
Social support	−0.161	0.032	−0.334	<0.001
Sense of coherence	−0.081	0.025	−0.231	0.001

*R^2^*: 0.52; source: self made.

**Table 4 ijerph-17-05230-t004:** Multiple linear regression by steps (dependent variable: anxiety).

Variables	Beta	Standard Error	Beta Standardised	*p*
Perceived burden	1.052	0.209	0.273	<0.001
Self-efficacy in care	−0.329	0.160	−0.107	0.041
Social support	−0.234	0.072	−0.217	0.001
Sense of coherence	−0.262	0.056	−0.333	<0.001

*R^2^*: 0.58; source: self made.

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
