# Peer review of "Personal and Family Resources Related to Depressive and Anxiety Symptoms in Women during Puerperium"

_ijerph, 2020, doi:10.3390/ijerph17145230_

Round 1

Reviewer 1 Report

This paper addresses an important and to a large extent overlooked phenomenon in today's puerperium care.

Personally, in the introduction I have missed the scientific findings that evidence the implications of maternal anxiety and depression in their interactions with the newborn (e.g. the Corinna Reck or Ruth Feldman’s Team studies), and the consequences on emotional regulation, attachment system’s formation and child development.

In the same vein, in the introduction and discussion sections I miss one of the factors associated with depression and postpartum anxiety that is gaining much importance as a mediator of the variables selected in this study: maternal attachment(see systematic review by Warfa et al. 2014 ).

In this sense, the selection of the “sense of coherence” shows a step from a pathogenic to a salutogenic way of view, adding and taking into account other variables such as well-being, close relationships and intimacy (all of which are closely related to each other). Therefore, as a recommendation for future studies, it would be interesting to provide this perspective.

In relation to stressful life events, it can also be interesting to know if they occurred before adulthood. Epidemiological studies in relation to Adverse Childhood Experiences (ACE) are worthy of consideration.

Regarding the methodology, two points are not clear to me:

1 - the way the "random sampling" was done, how many cases came in, how many were chosen, how many were rejected ... etc?

2 - If the averages of the sociodemographic variables of the sample ("Marital Status", Educational level, Family Income ....) are comparable to those of the population of Jaen or Spain.

Minor comments

Line 228 - Table 1 - Statistics like "n" and "M" conventionally go in italics

                 - Table 1 - DE should be SD (italics also)

                - Table 1 - IC should be CI

               - Table 1 - subtitles as "marital status) without bold

Line 230 - Talble 1 - Set the decimals of the results to 2

Line 257 - Variable dependent should be symptoms of anxiety

References:

Mittelmark, M. B., Sagy, S., Eriksson, M., Bauer, G. F., Pelikan, J. M., Lindström, B., & Espnes, G. A. (Eds.). (2017). The Handbook of Salutogenesis. CFS Courier Forschungsinstitut Senckenberg. Cham: Springer International Publishing. https://doi.org/10.1007/978-3-319-04600-6

Warfa, N., Harper, M., Nicolais, G., & Bhui, K. (2014). Adult attachment style as a risk factor for maternal postnatal depression: A systematic review. BMC Psychology, 2(1). https://doi.org/10.1186/s40359-014-0056-x

Reviewer 2 Report

This paper adds research on the psychological disorders, as depressive symptoms and anxiety, suffered by women in the puerperium period. Puerperal depression is considered a mild mental and behavioural disorder. The prevalence of puerperal depression is determined through questionnaires self-filled anonymously during the 5th week after the delivery, with a descriptive and cross-sectional design.

it's not clear how the questionnaires overestimate the presence of the disorders. In the discussion, the authors find that increased depressive symptoms and anxiety correlate with less perceived social support and greater perceived burden. Also with less self-efficacy in care and a lesser sense of coherence.

Recommendations naturally arise in the sense of improving the perception of competence in care, the sensation of feeling connected or the perception of new born rearing. The degree of subjectivity of these results could be considered too excessive. Extending the research to more objective variables should be considered. More statistical analysis and significant information could be drawn from variables already included in the study.

Reviewer 3 Report

Thank you for the opportunity to review this paper, I think it is a good piece of work. I do have some comments which can hopefully help improve it.

The introduction is lacking a lot of depth and review of the previous literature to build the rationale for the study.

Line#124 What was the idea behind calculating sample for r=0.175? Was it a pilot observation or data from literature? Please clarify or add a reference.

Line #205 Confidence interval is not a measure of central tendency. By using “typical deviation”, you meant here standard deviation? Please revise and use consistent terminology to avoid confusion from readers. Also, it would be best if you decide what measure of variability you present (Regards Table 1). Providing confidence interval makes more sense if your study aims to provide an estimate for the population. As long as, you want to summarize the group characteristics, it’s better to show standard deviations.

Line #207 Similarly for percentages, due to the aim of your analysis, I would not present Cis for qualitative variables

Line #209-210, 213 You name analysis steps which were followed if normality assumption was not met. But previously regarding descriptive statistics, you did not mention the use of, e.g. median and interquartile range to describe those variables. Otherwise, summarizing non-normal data with mean and 95%CI OR SD is a mistake.

Line #205-216 Regarding the previous point, you should name the method you used to check the normality of data (i.e. Shapiro-Wilk test, visual inspection of histogram or QQ plot)

Line #210 As you dichotomize EPDS scale (10 as depression) it would make more sense to use logistic regression with binary outcome depression/no depression.

Line #211 It is not clear how your regression models are built. You name use of a stepwise selection of covariates which has to be treated with caution. The problem is that stepwise excludes variables above certain p-value threshold. Your models did not consider common risk factors for postpartum depression named in the literature (e.g. age, number of prior pregnancies), which can leave place for residual confounding. Please name a list of candidate variables for the analysis or consider running models with variable selection based on your expert knowledge.

Line #212 I would drop information on assumption checking for regression analysis. This is an obvious step.

Line #208, 214 Please be careful about the use and meaning of “bi-“ and “multivariate” terms. As the term “bivariate” can be used to name correlations, but I would omit that. Use of the term “multivariate” is a mistake. In public health research, “multivariable analysis” is commonly interchanged with “multivariate analysis”, which are two distinct issues. Multivariate analysis in terms of regression is a regression with multiple DEPENDENT variables, whereas the multivariable model means the inclusion of multiple INDEPENDENT variables. Please rename.

Thank you for considering my comments and I look forward to seeing the revised version.

Round 2

Reviewer 3 Report

The authors carefully responded to all comments and corrected some
mistakes.